# Linear Codes Constructed from Two Weakly Regular Plateaued Functions with Index (*p* − 1)/2

**DOI:** 10.3390/e26060455

**Published:** 2024-05-27

**Authors:** Shudi Yang, Tonghui Zhang, Zheng-an Yao

**Affiliations:** 1School of Mathematical Sciences, Qufu Normal University, Jining 273165, China; zhangthvvs@126.com; 2School of Mathematics and Statistics, Fujian Normal University, Fuzhou 350117, China; 3School of Mathematics, Sun Yat-sen University, Guangzhou 510275, China; mcsyao@mail.sysu.edu.cn

**Keywords:** linear code, weight distribution, Walsh transform, plateaued function

## Abstract

Linear codes are the most important family of codes in cryptography and coding theory. Some codes only have a few weights and are widely used in many areas, such as authentication codes, secret sharing schemes and strongly regular graphs. By setting p≡1(mod4), we constructed an infinite family of linear codes using two distinct weakly regular unbalanced (and balanced) plateaued functions with index (p−1)/2. Their weight distributions were completely determined by applying exponential sums and Walsh transform. As a result, most of our constructed codes have a few nonzero weights and are minimal.

## 1. Introduction

Let *p* be a prime number and $F_{p}$ the finite field with *p* elements. We denote *C* to be a linear code over $F_{p}$ with parameters [n,k,d], which that means *C* is a subspace of dimension *k* with minimum distance *d* of the vector space $F_{p}^{n}$. Compared with nonlinear codes, linear codes are easier to describe, encode and decode, due to their algebraic structure, so they have many applications in cryptography and communications. See [1] for more information about linear codes.

For a codeword $c=(c_{0},c_{1},\ldots ,c_{n-1})\in C$, its weight is defined by
$$\begin{array}{c} \mathrm{wt}\left(c\right)=\#\left\{0⩽i<n:c_{i}\neq 0\right\}. \end{array}$$
Then, the weight distribution of *C* is the sequence $\left(A_{0},A_{1},A_{2},\ldots ,A_{n}\right)$, where $A_{0}=1$ and $A_{w}$ stands for the number of codewords in *C* that have weight *w*, for 0⩽w⩽n, i.e.,
$$\begin{array}{c} A_{w}=\#\left\{c\in C:\mathrm{wt}\left(c\right)=w\right\}. \end{array}$$
The code *C* is called *t*-weight if the number of nonzero $A_{w}$ for 1⩽w⩽n equals *t*. Linear codes with a few nonzero weights have attracted much attention in recent decades due to their wide applications in theory and practice, see [2,3,4,5,6,7,8,9,10,11]. Some linear codes are constructed from bent functions [6,12], square functions [13] and weakly regular plateaued functions [3,5,7].

In what follows, we always assume *p* is an odd prime. Now, let us introduce an efficient way to construct linear codes, which was proposed by Ding et al. [14]. Let $q=p^{m}$ and *D* be a subset of $F_{q}$ of size *n*. We define
$$\begin{array}{c} C_{D}=\left\{c\left(a\right)={\left(\mathrm{Tr}(ax)\right)}_{x\in D}:a\in F_{q}\right\}, \end{array}$$
where Tr is the absolute trace function. It can be checked that $C_{D}$ is a linear code of length *n*. The set *D* is called the defining set of $C_{D}$. This approach was generalized by Li et al. [15], who defined a class of codes by
(1)$$\begin{array}{c} C_{D}=\left\{c\left(a,b\right)={\left(\mathrm{Tr}(ax+by)\right)}_{(x,y)\in D}:a,b\in F_{q}\right\}, \end{array}$$
where the defining set *D* is a subset of $F_{q}^{2}$. Let $c\in F_{p}$. For *p*-ary functions *f* and *g*, we define
$$\begin{array}{c} D\left(c\right)=\left\{\left(x,y\right)\in F_{q}^{2}\setminus \left\{\left(0,0\right)\right\}:f\left(x\right)+g\left(y\right)=c\right\}. \end{array}$$
Based on [15], Wu et al. [16] offered new linear codes using the defining set D(0), where *f* and *g* are weakly regular bent functions from $F_{q}$ to $F_{p}$. Later, Cheng et al. in [3] introduced several linear codes $C_{D(0)}$ of (Equation 1) with a few weights by considering *f* and *g* to be weakly regular unbalanced *s*-plateaued functions in the defining set D(0), where 0⩽s⩽m. In 2022, Sınak [17] went deeper by choosing the weakly regular unbalanced and balanced $s_{f}$-plateaued function *f* and $s_{g}$-plateaued function *g* in D(0), where $0⩽s_{f},\;s_{g}⩽m$. Very recently, Yang et al. [18] continued the research of [17] by considering two weakly regular balanced plateaued functions in the defining set D(c), where c≠0. All of them studied the indexes of *f* and *g* among the set {2,p−1}, that is, $l_{f},l_{g}\in \left\{2,p-1\right\}$.

Along this research line, we further consider the index of (p−1)/2, where p≡1(mod4). Let *f* and *g* be certain weakly regular unbalanced and balanced *s*-plateaued and *t*-plateaued functions, respectively, for 0⩽s,t⩽m. The defining set is denoted by
(2)$$\begin{array}{c} D_{f,g}=\left\{\left(x,y\right)\in F_{q}^{2}\setminus \left\{\left(0,0\right)\right\}:f\left(x\right)+g\left(y\right)=0\right\}. \end{array}$$
For clarity, we only concentrate on the case of $l_{g}=\left(p-1\right)/2$ and $l_{f}\in \left\{2,p-1\right\}$, since the case of $l_{f}=\left(p-1\right)/2$ and $l_{g}\in \left\{2,p-1\right\}$ will lead to similar results (also, see Remark 3 for the case of $l_{f}=l_{g}=\left(p-1\right)/2$). In this paper, we consider the constructed codes $C_{D_{f,g}}$ of (Equation 1) and (Equation 2). In detail, we will completely determine their weight distributions using the theory of exponential sums and Walsh transform.

The rest of this paper is arranged as follows. We first present, in Section 2, an introduction to the mathematical foundations. Section 3 gives necessary results for our computation. Our main results are proposed in Section 4, where we study the weight distributions and the parameters of our constructed codes and their punctured ones. Section 5 shows the minimality and applications of these codes. Finally, the whole paper is concluded in Section 6.

## 2. Mathematical Background

In this section, let us have a quick glance at the mathematical background, including cyclotomic classes, cyclotomic fields, the theory of exponential sums and weakly regular plateaued functions. We recall that $q=p^{m}$ and m⩾2. We denote by $S_{q}$ (resp. $N_{sq}$) the set of square (resp. non-square) elements in $F_{p}^{*}$.

### 2.1. Cyclotomic Classes and Cyclotomic Fields

Let θ be a fixed primitive element of $F_{q}$ and N⩾2 be a divisor of q−1. For 0⩽i<N, the *i*-th cyclotomic classes of order *N* are defined by $C_{i}^{\left(N,q\right)}=\theta^{i}\langle \theta^{N}\rangle$, where $\langle \theta^{N}\rangle$ stands for the subgroup generated by $\theta^{N}$.

The *p*-th cyclotomic field is denoted by $K=Q(\zeta_{p})$, where $\zeta_{p}=\exp\left(\frac{2\pi \sqrt{-1}}{p}\right)$. From [19], we know that the Galois group Gal(K/Q) is given by $\left\{\sigma_{z}:z\in F_{p}^{*}\right\}$, where the automorphism $\sigma_{z}$ of *K* is defined by $\sigma_{z}\left(\zeta_{p}\right)=\zeta_{p}^{z}$. Let η be the quadratic character of $F_{p}$. Then, $\sigma_{z}\left(\sqrt{p^{*}}\right)=\eta \left(z\right)\sqrt{p^{*}}$, where $p^{*}=\eta \left(-1\right)p$.

### 2.2. Exponential Sums

We denote by $\eta_{m}$ the quadratic character of $F_{q}$, where $q=p^{m}$. Let $G(\eta_{m})$ be the quadratic Gauss sum over $F_{q}$ defined by
$$\begin{array}{c} G\left(\eta_{m}\right)=\sum_{x\in F_{q}^{*}}\eta_{m}\left(x\right)\chi_{1}\left(x\right), \end{array}$$
where $\chi_{1}\left(x\right)=\zeta_{p}^{\mathrm{Tr}(x)}$ is the canonical additive character, and Tr is the absolute trace function. It is well known that $G\left(\eta_{m}\right)={\left(-1\right)}^{m-1}{\sqrt{p^{*}}}^{m}$ and $G\left(\eta \right)=\sqrt{p^{*}}$.

For n∈N and $a\in F_{q}^{*}$, the Jacobsthal sum is defined by
$$\begin{array}{c} H_{n}\left(a\right)=\sum_{x\in F_{q}}\eta_{m}\left(x^{n+1}+ax\right)=\sum_{x\in F_{q}}\eta_{m}\left(x\right)\eta_{m}\left(x^{n}+a\right). \end{array}$$
We define
$$\begin{array}{c} I_{n}\left(a\right)=\sum_{x\in F_{q}}\eta_{m}\left(x^{n}+a\right). \end{array}$$
It is a companion sum related to Jacobsthal sums because $I_{2n}\left(a\right)=I_{n}\left(a\right)+H_{n}\left(a\right)$, which is due to Theorem 5.50 in [20]. We can evaluate easily that $I_{1}\left(a\right)=0$ and $I_{2}\left(a\right)=-1$ for all $a\in F_{q}^{*}$. In general, the sums $I_{n}\left(a\right)$ can be described in terms of Jacobi sums.

**Lemma** **1**(Theorem 5.51, [20])**.**
*For all $a\in F_{q}^{*}$ and n∈N, we have*
$$I_{n}\left(a\right)=\eta_{m}\left(a\right)\sum_{j=1}^{d-1}\lambda^{j}\left(-a\right)J\left(\lambda^{j},\eta_{m}\right),$$
*where λ is a multiplicative character of $F_{q}$ of order d=gcd(n,q−1), and $J(\lambda^{j},\eta_{m})$ is a Jacobi sum in $F_{q}$.*

**Lemma** **2**(Theorem 5.33, [20])**.**
*Let $q=p^{m}$ be odd and $f\left(x\right)=a_{2}x^{2}+a_{1}x+a_{0}\in F_{q}\left[x\right]$ with $a_{2}\neq 0$. Then,*
$$\sum_{x\in F_{q}}\zeta_{p}^{\mathrm{Tr}(f(x))}=\zeta_{p}^{\mathrm{Tr}(a_{0}-a_{1}^{2}{\left(4a_{2}\right)}^{-1})}\eta_{m}\left(a_{2}\right)G\left(\eta_{m}\right).$$

### 2.3. Weakly Regular Plateaued Functions

Let $f:F_{q}\to F_{p}$ be a *p*-ary function. For $\beta \in F_{q}$, the Walsh transform of *f* is defined by
$$\begin{array}{c} {\hat{\chi }}_{f}\left(\beta \right)=\sum_{x\in F_{q}}\zeta_{p}^{f(x)-\mathrm{Tr}(\beta x)}. \end{array}$$
A function *f* is said to be balanced if ${\hat{\chi }}_{f}\left(0\right)=0$; otherwise, it is said to be unbalanced.

Plateaued functions in characteristic 2 were first studied by Zheng et al. [21] for cryptographic applications in 1999, and later in any general characteristic *p* by Mesnager [22] in 2014. Several years ago, Mesnager et al. presented the definition of (non-)weakly regular plateaued functions in their work [23]. We follow the notation used in [23]. A function *f* is *s*-plateaued if $|{\hat{\chi }}_{f}{\left(\beta \right)|}^{2}\in \left\{0,p^{m+s}\right\}$ for each $\beta \in F_{q}$, where 0⩽s⩽m. Let $S_{f}$ be the Walsh support of *f*. In fact,
$$\begin{array}{c} S_{f}=\{\beta \in F_{q}:|{\hat{\chi }}_{f}\left(\beta \right){|}^{2}=p^{m+s}\}. \end{array}$$
According to [22], the cardinality of $S_{f}$ is given by $\#S_{f}=p^{m-s}$.

**Definition** **1**([23])**.**
*A function f is called weakly regular s-plateaued if there exists a complex number u, |u|=1, such that*
$$\begin{array}{c} {\hat{\chi }}_{f}\left(\beta \right)\in \left\{0,up^{\frac{m+s}{2}}\zeta_{p}^{g(\beta )}\right\} \end{array}$$
*for all $\beta \in F_{q}$, where g is a p-ary function over $F_{q}$ satisfying g(β)=0 for all $\beta \in F_{q}\setminus S_{f}$. Otherwise, if u depends on β, then f is called non-weakly regular s-plateaued.*

**Lemma** **3**(Lemma 5, [23])**.**
*Let $\beta \in F_{q}$ and f a weakly regular s-plateaued function. For every $\beta \in S_{f}$, we have*
$$\begin{array}{c} {\hat{\chi }}_{f}\left(\beta \right)=\varepsilon_{f}{\sqrt{p^{*}}}^{m+s}\zeta_{p}^{f^{★}\left(\beta \right)}, \end{array}$$
*where $\varepsilon_{f}\in \left\{\pm 1\right\}$ is the sign of ${\hat{\chi }}_{f}$ and $f^{★}$ is a p-ary function over $F_{q}$ with $f^{★}\left(\beta \right)=0$ for all $\beta \in F_{q}\setminus S_{f}$. We call $f^{★}$ the dual function of f.*

In 2020, Mesnager and Sınak [5,7] defined two subclasses of weakly regular plateaued functions.

**Definition** **2**([5,7])**.**
*Let f be a weakly regular unbalanced (resp. balanced) s-plateaued function with 0⩽s⩽m. We denote by* WRP *(resp.* WRPB*) the subclass of the unbalanced (resp. balanced) functions f that meet the following homogeneous conditions simultaneously:*
*1.* *f(0)=0;**2.* *There exists a positive integer $h_{f}$, such that $2|h_{f}$, $\gcd(h_{f}-1,p-1)=1$ and $f\left(zx\right)=z^{h_{f}}f\left(x\right)$ for every $z\in F_{p}^{*}$.*

**Remark** **1.**
*It is clear that $0\in S_{f}$ (resp. $0\notin S_{f}$) whenever f∈WRP (resp. f∈WRPB).*


The following lemmas, due to [5,17], play a significant role in the following calculation.

**Lemma** **4**(Lemma 6, [5])**.**
*Let f∈WRP or f∈WRPB with ${\hat{\chi }}_{f}\left(\beta \right)=\varepsilon_{f}{\sqrt{p^{*}}}^{m+s}\zeta_{p}^{f^{★}\left(\beta \right)}$, where $\beta \in S_{f}$. Then, for $z\in F_{p}^{*}$, we have $z\beta \in S_{f}$ if $\beta \in S_{f}$, and otherwise, we have $z\beta \in F_{q}\setminus S_{f}$.*

**Lemma** **5**(Propositions 2 and 3, [5])**.**
*Let f∈WRP or f∈WRPB with ${\hat{\chi }}_{f}\left(\beta \right)=\varepsilon_{f}{\sqrt{p^{*}}}^{m+s}\zeta_{p}^{f^{★}\left(\beta \right)}$, where $\beta \in S_{f}$. Then, $f^{★}\left(0\right)=0$ and $f^{★}\left(z\beta \right)=z^{l_{f}}f^{★}\left(\beta \right)$ for all $z\in F_{p}^{*}$, where $2|l_{f}$ and $\gcd(l_{f}-1,p-1)=1$. We call $l_{f}$ the index of f.*

**Remark** **2.**
*According to Lemma 5, if we take $l_{f}=\left(p-1\right)/2$, then we must have p≡1(mod4).*


**Lemma** **6**(Lemma 10, [5])**.**
*Let f∈WRP or f∈WRPB with ${\hat{\chi }}_{f}\left(\beta \right)=\varepsilon_{f}{\sqrt{p^{*}}}^{m+s}\zeta_{p}^{f^{★}\left(\beta \right)}$, where $\beta \in S_{f}$. For $c\in F_{p}$, we define*
$$\begin{array}{c} N_{f}\left(c\right)=\#\left\{\beta \in S_{f}:f^{★}\left(\beta \right)=c\right\}. \end{array}$$
*When 2∣m−s,*
$$\begin{array}{c} N_{f}\left(c\right)=\left\{\begin{array}{cc} p^{m-s-1}+\left(p-1\right)\eta^{m+1}\left(-1\right)\varepsilon_{f}{\sqrt{p^{*}}}^{m-s-2}, & \mathrm{if}\;c=0,\\ p^{m-s-1}-\eta^{m+1}\left(-1\right)\varepsilon_{f}{\sqrt{p^{*}}}^{m-s-2}, & \mathrm{if}\;c\neq 0. \end{array}\right. \end{array}$$
*Otherwise,*
$$\begin{array}{c} N_{f}\left(c\right)=\left\{\begin{array}{cc} p^{m-s-1}, & \mathrm{if}\;c=0,\\ p^{m-s-1}+\eta \left(c\right)\eta^{m}\left(-1\right)\varepsilon_{f}{\sqrt{p^{*}}}^{m-s-1}, & \mathrm{if}\;c\neq 0. \end{array}\right. \end{array}$$

**Lemma** **7**(Lemma 3.12, [17])**.**
*Let f,g∈WRP or f,g∈WRPB with ${\hat{\chi }}_{f}\left(\alpha \right)=\varepsilon_{f}{\sqrt{p^{*}}}^{m+s}\zeta_{p}^{f^{★}\left(\alpha \right)}$ and ${\hat{\chi }}_{g}\left(\beta \right)=\varepsilon_{g}{\sqrt{p^{*}}}^{m+t}\zeta_{p}^{g^{★}\left(\beta \right)}$, where $\alpha \in S_{f}$ and $\beta \in S_{g}$. We define*
$$\begin{array}{cc} \; & T\left(0\right)=\#\{\left(a,b\right)\in S_{f}\times S_{g}:f^{★}\left(a\right)+g^{★}\left(b\right)=0\},\\ \; & T\left(c\right)=\#\left\{\left(a,b\right)\in S_{f}\times S_{g}:f^{★}\left(a\right)+g^{★}\left(b\right)=c\right\}\mathrm{for}\;c\in F_{p}^{*}. \end{array}$$
*Then, we have*
$$\begin{array}{cc} \; & T\left(0\right)=\left\{\begin{array}{cc} p^{2m-s-t-1}+\left(p-1\right)p^{-1}\varepsilon_{f}\varepsilon_{g}{\sqrt{p^{*}}}^{2m-s-t}, & \mathrm{if}2|s+t,\\ p^{2m-s-t-1}, & \mathrm{if}2∤s+t, \end{array}\right.\\ \; & T\left(c\right)=\left\{\begin{array}{cc} p^{2m-s-t-1}-p^{-1}\varepsilon_{f}\varepsilon_{g}{\sqrt{p^{*}}}^{2m-s-t}, & \mathrm{if}2|s+t,\\ p^{2m-s-t-1}+\eta \left(c\right)\varepsilon_{f}\varepsilon_{g}{\sqrt{p^{*}}}^{2m-s-t-1}, & \mathrm{if}2∤s+t. \end{array}\right. \end{array}$$

**Lemma** **8**(Lemma 3.7, [17])**.**
*We write $n=\#D_{f,g}$, where $D_{f,g}$ is defined by (Equation 2) and f,g are given in Lemma 7. If f,g∈WRPB, then $n=p^{2m-1}-1$. If f,g∈WRP, then*
$$\begin{array}{c} n=\left\{\begin{array}{cc} p^{2m-1}-1, & \mathrm{if}2∤s+t,\\ p^{2m-1}-1+\left(p-1\right)p^{-1}\varepsilon_{f}\varepsilon_{g}{\sqrt{p^{*}}}^{2m+s+t}, & \mathrm{if}2|s+t. \end{array}\right. \end{array}$$

## 3. Auxiliary Results

To ensure that the frequency of each weight appears in our codes, we will need the following lemmas.

**Lemma** **9.**
*Let p≡1(mod2). For the quadratic character η over $F_{p}$, we have*

$$\begin{array}{cc} \sum_{u\in S_{q}}\sum_{\begin{array}{c} v\in S_{q}\\ v\neq \pm u \end{array}}\eta \left(u+v\right) & =-\frac{p-1}{2}\left(\eta \left(2\right)+1\right),\\ \sum_{u\in N_{sq}}\sum_{\begin{array}{c} v\in N_{sq}\\ v\neq \pm u \end{array}}\eta \left(u+v\right) & =\frac{p-1}{2}\left(\eta \left(2\right)+1\right). \end{array}$$



**Proof.** We note that $-1\in S_{q}$ if p≡1(mod4), and otherwise, $-1\in N_{sq}$ if p≡3(mod4). Thus,
$$\begin{array}{cc} \sum_{u\in S_{q}}\sum_{\begin{array}{c} v\in S_{q}\\ v\neq \pm u \end{array}}\eta \left(u+v\right) & =\sum_{u\in S_{q}}\eta \left(u\right)\sum_{\begin{array}{c} v\in S_{q}\\ v\neq \pm u \end{array}}\eta \left(1+\frac{v}{u}\right)\\ \; & =\sum_{u\in S_{q}}\sum_{\begin{array}{c} v\in S_{q}\\ v\neq \pm 1 \end{array}}\eta \left(1+v\right)\\ \; & =\frac{p-1}{2}\left(\sum_{v\in S_{q}}\eta \left(1+v\right)-\eta \left(2\right)\right)\\ \; & =\frac{p-1}{2}\left(\frac{1}{2}\sum_{x\in F_{p}}\eta \left(1+x^{2}\right)-\frac{1}{2}-\eta \left(2\right)\right)\\ \; & =\frac{p-1}{2}\left(\frac{1}{2}I_{2}\left(1\right)-\frac{1}{2}-\eta \left(2\right)\right). \end{array}$$
The first assertion then follows from $I_{2}\left(1\right)=-1$. The second one is analogously proved and is omitted here. □

**Lemma** **10.**
*Let p≡1(mod4) and f,g be given as Lemma 7. We suppose that s+t is odd. We write γ=2m−s−t and*

$$\begin{array}{cc} B_{S_{q}} & =\#\{\left(a,b\right)\in S_{f}\times S_{g}:f^{★}\left(a\right)+g^{★}\left(b\right)\in S_{q},f^{★}\left(a\right)-g^{★}\left(b\right)\in S_{q}\},\\ B_{N_{sq}} & =\#\{\left(a,b\right)\in S_{f}\times S_{g}:f^{★}\left(a\right)+g^{★}\left(b\right)\in N_{sq},f^{★}\left(a\right)-g^{★}\left(b\right)\in N_{sq}\}. \end{array}$$

*Then, if 2∤m−s and 2∣m−t, we have*

BSq=p−12pγ−3(p−12pγ−1−η(2)εfpm−t==+p+12εgpm−s−1+(η(2)+p)εfεg),BNsq=p−12pγ−3(p−12pγ−1+η(2)εfpm−t==+p+12εgpm−s−1−(η(2)+p)εfεg).

*Otherwise, if 2∣m−s and 2∤m−t, we have*

BSq=p−12pγ−3(p−12pγ−1−η(2)εgpm−s==+p+12εfpm−t−1+(η(2)+p)εfεg),BNsq=p−12pγ−3(p−12pγ−1+η(2)εgpm−s==+p+12εfpm−t−1−(η(2)+p)εfεg).



**Proof.** We only calculate $B_{S_{q}}$ for the case 2∤m−s and 2∣m−t. Let $f^{★}\left(a\right)+g^{★}\left(b\right)=u$, $f^{★}\left(a\right)-g^{★}\left(b\right)=v$, where $u,v\in F_{p}^{*}$. So, $f^{★}\left(a\right)=\frac{u+v}{2}$, $g^{★}\left(b\right)=\frac{u-v}{2}$ and consequently,
$$\begin{array}{c} B_{S_{q}}=\sum_{u\in S_{q}}\sum_{v\in S_{q}}N_{f}\left(\frac{u+v}{2}\right)N_{g}\left(\frac{u-v}{2}\right), \end{array}$$
where $N_{f}$ and $N_{g}$ are computed in Lemma 6. It follows that
$$\begin{array}{cc} B_{S_{q}} & =\sum_{u\in S_{q}}N_{f}\left(u\right)N_{g}\left(0\right)+\sum_{u\in S_{q}}N_{f}\left(0\right)N_{g}\left(u\right)+S, \end{array}$$
where
(3)$$S=\sum_{u\in S_{q}}\sum_{\begin{array}{c} v\in S_{q}\\ v\neq \pm u \end{array}}N_{f}\left(\frac{u+v}{2}\right)N_{g}\left(\frac{u-v}{2}\right).$$
We observe that $\frac{u-v}{2}\neq 0$ in (Equation 3). If we write $c=\frac{u-v}{2}\neq 0$, then, from Lemma 6,
$$\begin{array}{cc} S & =N_{g}\left(c\right)\sum_{u\in S_{q}}\sum_{\begin{array}{c} v\in S_{q}\\ v\neq \pm u \end{array}}N_{f}\left(\frac{u+v}{2}\right)\\ \; & =N_{g}\left(c\right)\sum_{u\in S_{q}}\sum_{\begin{array}{c} v\in S_{q}\\ v\neq \pm u \end{array}}\left(p^{m-s-1}+\eta \left(\frac{u+v}{2}\right)\varepsilon_{f}{\sqrt{p}}^{m-s-1}\right)\\ \; & =N_{g}\left(c\right)\left(\frac{p-1}{2}\cdot \frac{p-5}{2}p^{m-s-1}+\eta \left(2\right)\varepsilon_{f}{\sqrt{p}}^{m-s-1}\sum_{u\in S_{q}}\sum_{\begin{array}{c} v\in S_{q}\\ v\neq \pm u \end{array}}\eta \left(u+v\right)\right). \end{array}$$
The desired assertion then follows from Lemmas 6 and 9. □

## 4. Main Results

In this section, we will give our main results of the weight distributions of the desired linear codes $C_{D_{f,g}}$ defined by (Equation 1) and (Equation 2). Let us fix some notation that will be used throughout this section. Let p≡1(mod4) and f,g∈WRP or f,g∈WRPB. For each $\alpha \in S_{f}$ and $\beta \in S_{g}$, we may assume from Lemma 3 that ${\hat{\chi }}_{f}\left(\alpha \right)=\varepsilon_{f}{\sqrt{p}}^{m+s}\zeta_{p}^{f^{★}\left(\alpha \right)}$ and ${\hat{\chi }}_{g}\left(\beta \right)=\varepsilon_{g}{\sqrt{p}}^{m+t}\zeta_{p}^{g^{★}\left(\beta \right)}$, where $\varepsilon_{f},\varepsilon_{g}\in \left\{\pm 1\right\}$ and 0⩽s,t⩽m. The indexes of *f* and *g* are $l_{f}$ and $l_{g}$ such that $l_{f}\in \left\{2,p-1\right\}$ and $l_{g}=\left(p-1\right)/2$.

For $\left(a,b\right)\in F_{q}^{2}\setminus \left\{\left(0,0\right)\right\}$, we define
(4)$$\begin{array}{c} N_{0}=\#\left\{\left(x,y\right)\in F_{q}^{2}:\mathrm{Tr}\left(ax+by\right)=0,f\left(x\right)+g\left(y\right)=0\right\}. \end{array}$$
In what follows, we always denote γ=2m−s−t and τ=2m+s+t for abbreviation purposes.

### 4.1. The Calculation of $N_{0}$

The values of $N_{0}$ in (Equation 4) are stated in Lemmas 11–13.

**Lemma** **11.**
*Let f,g∈WRP or f,g∈WRPB with $l_{g}=\left(p-1\right)/2$. We suppose that 2∤s+t and (a,b)≠(0,0). We always have $N_{0}=p^{2m-2}$ if $\left(a,b\right)\notin S_{f}\times S_{g}$. Otherwise, the following statements hold.*

*When $l_{f}=p-1$,*

$$\begin{array}{c} N_{0}=\left\{\begin{array}{cc} p^{2m-2}+\frac{p-1}{2}\eta \left(2\right)\varepsilon_{f}\varepsilon_{g}{\sqrt{p}}^{\tau -3}, & \mathrm{if}f^{★}\left(a\right)\in S_{q},g^{★}\left(b\right)=\pm f^{★}\left(a\right),\\ p^{2m-2}-\frac{p-1}{2}\eta \left(2\right)\varepsilon_{f}\varepsilon_{g}{\sqrt{p}}^{\tau -3}, & \mathrm{if}f^{★}\left(a\right)\in N_{sq},g^{★}\left(b\right)=\pm f^{★}\left(a\right),\\ p^{2m-2}+\left(p-1\right)\varepsilon_{f}\varepsilon_{g}{\sqrt{p}}^{\tau -3}, & \mathrm{if}f^{★}\left(a\right)+g^{★}\left(b\right)\in S_{q},f^{★}\left(a\right)-g^{★}\left(b\right)\in S_{q},\\ p^{2m-2}-\left(p-1\right)\varepsilon_{f}\varepsilon_{g}{\sqrt{p}}^{\tau -3}, & \mathrm{if}f^{★}\left(a\right)+g^{★}\left(b\right)\in N_{sq},f^{★}\left(a\right)-g^{★}\left(b\right)\in N_{sq},\\ p^{2m-2}, & \mathrm{otherwise}. \end{array}\right. \end{array}$$


*When $l_{f}=2$ and p≡1(mod8),*

$$\begin{array}{c} N_{0}=\left\{\begin{array}{cc} p^{2m-2}+\left(p-1\right)\varepsilon_{f}\varepsilon_{g}{\sqrt{p}}^{\tau -3}, & \mathrm{if}f^{★}\left(a\right)=0,g^{★}\left(b\right)\in \;S_{q}\\ \; & \mathrm{or}g^{★}\left(b\right)=0,f^{★}\left(a\right)\in S_{q},\\ p^{2m-2}-\left(p-1\right)\varepsilon_{f}\varepsilon_{g}{\sqrt{p}}^{\tau -3}, & \mathrm{if}f^{★}\left(a\right)=0,g^{★}\left(b\right)\in N_{sq}\\ \; & \mathrm{or}g^{★}\left(b\right)=0,f^{★}\left(a\right)\in N_{sq},\\ p^{2m-2}-2\left(p-1\right)\varepsilon_{f}\varepsilon_{g}{\sqrt{p}}^{\tau -3}, & \mathrm{if}f^{★}\left(a\right)\in S_{q},g^{★}\left(b\right)\in S_{q},\\ p^{2m-2}+2\left(p-1\right)\varepsilon_{f}\varepsilon_{g}{\sqrt{p}}^{\tau -3}, & \mathrm{if}f^{★}\left(a\right)\in N_{sq},g^{★}\left(b\right)\in N_{sq},\\ p^{2m-2}, & \mathrm{otherwise}. \end{array}\right. \end{array}$$


*When $l_{f}=2$ and p≡5(mod8),*

$$\begin{array}{c} N_{0}=\left\{\begin{array}{cc} p^{2m-2}, & \mathrm{if}f^{★}\left(a\right)=g^{★}\left(b\right)=0,\\ p^{2m-2}+\left(p-1\right)\varepsilon_{f}\varepsilon_{g}{\sqrt{p}}^{\tau -3}, & \mathrm{if}f^{★}\left(a\right)=0,g^{★}\left(b\right)\in \;S_{q}\\ \; & \mathrm{or}g^{★}\left(b\right)=0,f^{★}\left(a\right)\in S_{q},\\ p^{2m-2}-\left(p-1\right)\varepsilon_{f}\varepsilon_{g}{\sqrt{p}}^{\tau -3}, & \mathrm{if}f^{★}\left(a\right)=0,g^{★}\left(b\right)\in N_{sq}\\ \; & \mathrm{or}g^{★}\left(b\right)=0,f^{★}\left(a\right)\in N_{sq},\\ p^{2m-2}+\varepsilon_{f}\varepsilon_{g}{\sqrt{p}}^{\tau -3}\eta \left(f^{★}\left(a\right)\right)\left(I_{4}\left(\frac{g^{★}\left(b\right)}{f^{★}\left(a\right)}\right)-\eta \left(\frac{g^{★}\left(b\right)}{f^{★}\left(a\right)}\right)\right), & \mathrm{otherwise}, \end{array}\right. \end{array}$$

*where $I_{4}$ is a companion sum determined in Lemma 1.*


**Proof.** Let 2∤s+t. By Equation (Equation 4) and the orthogonal property of group characters,
(5)N0=1p2∑x,y∈Fq∑z∈Fpζpz(f(x)+g(y))∑h∈FpζphTr(ax+by)=1p2∑x,y∈Fq1+∑z∈Fp*ζpz(f(x)+g(y))1+∑h∈Fp*ζphTr(ax+by)=p2m−2+1p2∑z∈Fp*∑x,y∈Fqζpz(f(x)+g(y))==+1p2∑x,y∈Fq∑z∈Fp*∑h∈Fp*ζpz(f(x)+g(y))+hTr(ax+by)=p2m−2+p−2(Λ1+Λ2),
where we write
$$\begin{array}{cc} \Lambda_{1} & =\sum_{z\in F_{p}^{*}}\sum_{x,y\in F_{q}}{\zeta_{p}}^{z(f(x)+g(y))},\\ \Lambda_{2} & =\sum_{x,y\in F_{q}}\sum_{z\in F_{p}^{*}}\sum_{h\in F_{p}^{*}}{\zeta_{p}}^{z(f(x)+g(y))+h\mathrm{Tr}(ax+by)}. \end{array}$$
It follows that
$$\begin{array}{cc} \Lambda_{1} & =\sum_{z\in F_{p}^{*}}\sigma_{z}\left({\hat{\chi }}_{f}\left(0\right){\hat{\chi }}_{g}\left(0\right)\right)\\ \; & =\left\{\begin{array}{cc} 0, & \mathrm{if}\;f,g\in \mathrm{WRPB},\\ \varepsilon_{f}\varepsilon_{g}{\sqrt{p}}^{\tau }\sum_{z\in F_{p}^{*}}\eta^{s+t}\left(z\right), & \mathrm{if}\;f,g\in \mathrm{WRP}. \end{array}\right. \end{array}$$
So, we always have $\Lambda_{1}=0$ when 2∤s+t. Now, it is sufficient to determine $\Lambda_{2}$. We observe from its definition that
(6)$$\begin{array}{cc} \Lambda_{2} & =\sum_{z\in F_{p}^{*}}\sum_{h\in F_{p}^{*}}\sum_{x\in F_{q}}{\zeta_{p}}^{zf(x)-\mathrm{Tr}(hax)}\sum_{y\in F_{q}}{\zeta_{p}}^{zg(y)-\mathrm{Tr}(hby)}\\ \; & =\sum_{z\in F_{p}^{*}}\sum_{h\in F_{p}^{*}}\sum_{x\in F_{q}}{\zeta_{p}}^{z(f\left(x\right)-\mathrm{Tr}\left(\frac{h}{z}ax\right))}\sum_{y\in F_{q}}{\zeta_{p}}^{z(g\left(y\right)-\mathrm{Tr}\left(\frac{h}{z}by\right))}\\ \; & =\sum_{z\in F_{p}^{*}}\sum_{h\in F_{p}^{*}}\sigma_{z}\left({\hat{\chi }}_{f}\left(ha\right){\hat{\chi }}_{g}\left(hb\right)\right). \end{array}$$
Let $h\in F_{p}^{*}$. Obviously, when $\left(a,b\right)\notin S_{f}\times S_{g}$, $\left(ha,hb\right)\notin S_{f}\times S_{g}$ by Lemma 4. Hence, ${\hat{\chi }}_{f}\left(ha\right)=0$ or ${\hat{\chi }}_{g}\left(hb\right)=0$, and consequently, by (Equation 6),
$$\Lambda_{2}=0.$$
When $\left(a,b\right)\in S_{f}\times S_{g}$, then $\left(ha,hb\right)\in S_{f}\times S_{g}$. By (Equation 6), Lemmas 3 and 5, we obtain
(7)$$\begin{array}{cc} \Lambda_{2} & =\sum_{z\in F_{p}^{*}}\sigma_{z}\left(\sum_{h\in F_{p}^{*}}\varepsilon_{f}\varepsilon_{g}{\sqrt{p}}^{\tau }\zeta_{p}^{h^{l_{f}}f^{★}\left(a\right)+h^{l_{g}}g^{★}\left(b\right))}\right)\\ \; & =\varepsilon_{f}\varepsilon_{g}{\sqrt{p}}^{\tau }\sum_{z\in F_{p}^{*}}\eta^{s+t}\left(z\right)\sigma_{z}\left(\sum_{h\in F_{p}^{*}}\zeta_{p}^{h^{l_{f}}f^{★}\left(a\right)+h^{l_{g}}g^{★}\left(b\right)}\right)\\ \; & =\varepsilon_{f}\varepsilon_{g}{\sqrt{p}}^{\tau }\sum_{z\in F_{p}^{*}}\eta \left(z\right)\sigma_{z}\left(\sum_{h\in F_{p}^{*}}\zeta_{p}^{h^{l_{f}}f^{★}\left(a\right)+h^{l_{g}}g^{★}\left(b\right)}\right). \end{array}$$
In the following, we will determine $\Lambda_{2}$ in (Equation 7) by considering the cases of $l_{f}=p-1$ and $l_{f}=2$, separately.The first case is that $l_{f}=p-1$.In this case, $h^{p-1}=1$ for every $h\in F_{p}^{*}$. By (Equation 7), we have
$$\begin{array}{cc} \Lambda_{2} & =\varepsilon_{f}\varepsilon_{g}{\sqrt{p}}^{\tau }\sum_{z\in F_{p}^{*}}\eta \left(z\right)\sigma_{z}\left(\sum_{h\in S_{q}}\zeta_{p}^{f^{★}\left(a\right)+g^{★}\left(b\right)}+\sum_{h\in N_{sq}}\zeta_{p}^{f^{★}\left(a\right)-g^{★}\left(b\right)}\right)\\ \; & =\frac{p-1}{2}\varepsilon_{f}\varepsilon_{g}{\sqrt{p}}^{\tau }\left(\sum_{z\in F_{p}^{*}}\eta \left(z\right)\zeta_{p}^{z(f^{★}\left(a\right)+g^{★}\left(b\right))}+\sum_{z\in F_{p}^{*}}\eta \left(z\right)\zeta_{p}^{z(f^{★}\left(a\right)-g^{★}\left(b\right))}\right)\\ \; & =\left\{\begin{array}{cc} 0, & \mathrm{if}f^{★}\left(a\right)=g^{★}\left(b\right)=0,\\ \frac{p-1}{2}\varepsilon_{f}\varepsilon_{g}\eta \left(2f^{★}\left(a\right)\right){\sqrt{p}}^{\tau +1}, & \mathrm{if}f^{★}\left(a\right)=-g^{★}\left(b\right)\neq 0,\\ \frac{p-1}{2}\varepsilon_{f}\varepsilon_{g}\eta \left(2f^{★}\left(a\right)\right){\sqrt{p}}^{\tau +1}, & \mathrm{if}f^{★}\left(a\right)=g^{★}\left(b\right)\neq 0,\\ \frac{p-1}{2}\varepsilon_{f}\varepsilon_{g}\left(\eta \left(f^{★}\left(a\right)+g^{★}\left(b\right)\right)+\eta \left(f^{★}\left(a\right)-g^{★}\left(b\right)\right)\right){\sqrt{p}}^{\tau +1}, & \mathrm{otherwise}. \end{array}\right. \end{array}$$Now, let $l_{f}=2$; then, the proof is divided into two subcases.**Subcase (a)**: If p≡1(mod8), then $-1\in C_{0}^{\left(4,p\right)}$. So, from (Equation 7),
$$\begin{array}{cc} \Lambda_{2} & =\varepsilon_{f}\varepsilon_{g}{\sqrt{p}}^{\tau }\sum_{z\in F_{p}^{*}}\eta \left(z\right)\sigma_{z}\left(\sum_{h\in S_{q}}\zeta_{p}^{h^{2}f^{★}\left(a\right)+g^{★}\left(b\right)}+\sum_{h\in N_{sq}}\zeta_{p}^{h^{2}f^{★}\left(a\right)-g^{★}\left(b\right)}\right)\\ \; & =\varepsilon_{f}\varepsilon_{g}{\sqrt{p}}^{\tau }\sum_{z\in F_{p}^{*}}\eta \left(z\right)\sigma_{z}\left(\sum_{h\in S_{q}}\zeta_{p}^{h^{2}f^{★}\left(a\right)+g^{★}\left(b\right)}+\sum_{h\in N_{sq}}\zeta_{p}^{-(h^{2}f^{★}\left(a\right)+g^{★}\left(b\right))}\right)\\ \; & =\varepsilon_{f}\varepsilon_{g}{\sqrt{p}}^{\tau }\left(\sum_{z\in F_{p}^{*}}\eta \left(z\right)\sum_{h\in S_{q}}\zeta_{p}^{z(h^{2}f^{★}\left(a\right)+g^{★}\left(b\right))}+\sum_{z\in F_{p}^{*}}\eta \left(-z\right)\sum_{h\in N_{sq}}\zeta_{p}^{-z(h^{2}f^{★}\left(a\right)+g^{★}\left(b\right))}\right). \end{array}$$
Replacing −z by *z* in the last double sum above, we obtain from Lemma 2 that
$$\begin{array}{cc} \Lambda_{2}\; & =\varepsilon_{f}\varepsilon_{g}{\sqrt{p}}^{\tau }\sum_{z\in F_{p}^{*}}\eta \left(z\right)\sum_{h\in F_{p}^{*}}\zeta_{p}^{z(h^{2}f^{★}\left(a\right)+g^{★}\left(b\right))}\\ \; & =\varepsilon_{f}\varepsilon_{g}{\sqrt{p}}^{\tau }\sum_{z\in F_{p}^{*}}\eta \left(z\right)\zeta_{p}^{zg^{★}\left(b\right)}\sum_{h\in F_{p}^{*}}\zeta_{p}^{zh^{2}f^{★}\left(a\right)}\\ \; & =\left\{\begin{array}{cc} 0, & \mathrm{if}f^{★}\left(a\right)=g^{★}\left(b\right)=0,\\ \left(p-1\right)\varepsilon_{f}\varepsilon_{g}\eta \left(g^{★}\left(b\right)\right){\sqrt{p}}^{\tau +1}, & \mathrm{if}f^{★}\left(a\right)=0,g^{★}\left(b\right)\neq 0,\\ \left(p-1\right)\varepsilon_{f}\varepsilon_{g}\eta \left(f^{★}\left(a\right)\right){\sqrt{p}}^{\tau +1}, & \mathrm{if}f^{★}\left(a\right)\neq 0,g^{★}\left(b\right)=0,\\ -\left(p-1\right)\varepsilon_{f}\varepsilon_{g}\left(\eta \left(f^{★}\left(a\right)\right)+\eta \left(g^{★}\left(b\right)\right)\right){\sqrt{p}}^{\tau +1}, & \mathrm{otherwise}. \end{array}\right. \end{array}$$**Subcase (b)**: If p≡5(mod8), then $-1\in C_{2}^{\left(4,p\right)}$. So, from (Equation 7),
$$\begin{array}{cc} \Lambda_{2} & =\varepsilon_{f}\varepsilon_{g}{\sqrt{p}}^{\tau }\sum_{z\in F_{p}^{*}}\eta \left(z\right)\sigma_{z}\left(\sum_{h\in S_{q}}\zeta_{p}^{h^{2}f^{★}\left(a\right)+g^{★}\left(b\right)}+\sum_{h\in N_{sq}}\zeta_{p}^{h^{2}f^{★}\left(a\right)-g^{★}\left(b\right)}\right)\\ \; & =\varepsilon_{f}\varepsilon_{g}{\sqrt{p}}^{\tau }\sum_{z\in F_{p}^{*}}\eta \left(z\right)\sigma_{z}\left(\sum_{h\in S_{q}}\zeta_{p}^{h^{2}f^{★}\left(a\right)+g^{★}\left(b\right)}+\sum_{h\in S_{q}}\zeta_{p}^{-(h^{2}f^{★}\left(a\right)+g^{★}\left(b\right))}\right)\\ \; & =2\varepsilon_{f}\varepsilon_{g}{\sqrt{p}}^{\tau }\sum_{z\in F_{p}^{*}}\eta \left(z\right)\sigma_{z}\left(\sum_{h\in S_{q}}\zeta_{p}^{h^{2}f^{★}\left(a\right)+g^{★}\left(b\right)}\right)\\ \; & =2\varepsilon_{f}\varepsilon_{g}{\sqrt{p}}^{\tau }\sum_{h\in S_{q}}\sum_{z\in F_{p}^{*}}\eta \left(z\right)\zeta_{p}^{z(h^{2}f^{★}\left(a\right)+g^{★}\left(b\right))}. \end{array}$$We assume that $f^{★}\left(a\right)g^{★}\left(b\right)\neq 0$. If $\frac{g^{★}\left(b\right)}{f^{★}\left(a\right)}\in C_{2}^{\left(4,p\right)}$, then the equation $h^{2}f^{★}\left(a\right)+g^{★}\left(b\right)=0$ has exactly two solutions, $h_{1}$ and $h_{2}$, in $S_{q}$, where $h_{2}=-h_{1}$. Otherwise, if $\frac{g^{★}\left(b\right)}{f^{★}\left(a\right)}\notin C_{2}^{\left(4,p\right)}$, then the inequality $h^{2}f^{★}\left(a\right)+g^{★}\left(b\right)\neq 0$ holds for all *h* in $S_{q}$. Consequently, when $f^{★}\left(a\right)g^{★}\left(b\right)\neq 0$,
$$\begin{array}{cc} \Lambda_{2} & =2\varepsilon_{f}\varepsilon_{g}{\sqrt{p}}^{\tau +1}\sum_{h\in S_{q}}\eta \left(h^{2}f^{★}\left(a\right)+g^{★}\left(b\right)\right)\\ \; & =\varepsilon_{f}\varepsilon_{g}{\sqrt{p}}^{\tau +1}\sum_{h\in F_{p}^{*}}\eta \left(h^{4}f^{★}\left(a\right)+g^{★}\left(b\right)\right)\\ \; & =\varepsilon_{f}\varepsilon_{g}{\sqrt{p}}^{\tau +1}\eta \left(f^{★}\left(a\right)\right)\left(I_{4}\left(\frac{g^{★}\left(b\right)}{f^{★}\left(a\right)}\right)-\eta \left(\frac{g^{★}\left(b\right)}{f^{★}\left(a\right)}\right)\right), \end{array}$$
where $I_{4}$ is determined from Lemma 1. Thus, we conclude that
$$\begin{array}{c} \Lambda_{2}=\left\{\begin{array}{cc} 0, & \mathrm{if}f^{★}\left(a\right)=g^{★}\left(b\right)=0,\\ \left(p-1\right)\varepsilon_{f}\varepsilon_{g}\eta \left(g^{★}\left(b\right)\right){\sqrt{p}}^{\tau +1}, & \mathrm{if}f^{★}\left(a\right)=0,g^{★}\left(b\right)\neq 0,\\ \left(p-1\right)\varepsilon_{f}\varepsilon_{g}\eta \left(f^{★}\left(a\right)\right){\sqrt{p}}^{\tau +1}, & \mathrm{if}f^{★}\left(a\right)\neq 0,g^{★}\left(b\right)=0,\\ \varepsilon_{f}\varepsilon_{g}{\sqrt{p}}^{\tau +1}\eta \left(f^{★}\left(a\right)\right)\left(I_{4}\left(\frac{g^{★}\left(b\right)}{f^{★}\left(a\right)}\right)-\eta \left(\frac{g^{★}\left(b\right)}{f^{★}\left(a\right)}\right)\right), & \mathrm{otherwise}. \end{array}\right. \end{array}$$The desired conclusion then follows from (Equation 5), completing the proof. □

**Lemma** **12.**
*Let f,g∈WRP with $l_{g}=\left(p-1\right)/2$. We suppose that 2∣s+t and (a,b)≠(0,0). We always have $N_{0}=p^{2m-2}+\left(p-1\right)\varepsilon_{f}\varepsilon_{g}{\sqrt{p}}^{\tau -4}$ if $\left(a,b\right)\notin S_{f}\times S_{g}$. Otherwise, the following statements hold.*

*When lf=p−1, we have*

$$\begin{array}{c} N_{0}=\left\{\begin{array}{cc} p^{2m-2}+\left(p-1\right)\varepsilon_{f}\varepsilon_{g}{\sqrt{p}}^{\tau -2}, & \mathrm{if}f^{★}\left(a\right)=g^{★}\left(b\right)=0,\\ p^{2m-2}+\frac{p-1}{2}\varepsilon_{f}\varepsilon_{g}{\sqrt{p}}^{\tau -2}, & \mathrm{if}f^{★}\left(a\right)=-g^{★}\left(b\right)\neq \;0\\ \; & \mathrm{or}f^{★}\left(a\right)=g^{★}\left(b\right)\neq 0,\\ p^{2m-2}, & \mathrm{otherwise}. \end{array}\right. \end{array}$$


*When $l_{f}=2$ and p≡1(mod8), we have*

$$\begin{array}{c} N_{0}=\left\{\begin{array}{cc} p^{2m-2}+\left(p-1\right)\varepsilon_{f}\varepsilon_{g}{\sqrt{p}}^{\tau -2}, & \mathrm{if}f^{★}\left(a\right)=g^{★}\left(b\right)=0,\\ p^{2m-2}+2\varepsilon_{f}\varepsilon_{g}{\sqrt{p}}^{\tau -2}, & \mathrm{if}f^{★}\left(a\right)g^{★}\left(b\right)\in S_{q},\\ p^{2m-2}, & \mathrm{otherwise}. \end{array}\right. \end{array}$$


*When $l_{f}=2$ and p≡5(mod8), we have*

$$\begin{array}{c} N_{0}=\left\{\begin{array}{cc} p^{2m-2}+\left(p-1\right)\varepsilon_{f}\varepsilon_{g}{\sqrt{p}}^{\tau -2}, & \mathrm{if}f^{★}\left(a\right)=g^{★}\left(b\right)=0,\\ p^{2m-2}+4\varepsilon_{f}\varepsilon_{g}{\sqrt{p}}^{\tau -2}, & \mathrm{if}\frac{g^{★}\left(b\right)}{f^{★}\left(a\right)}\in C_{2}^{\left(4,p\right)},\\ p^{2m-2}, & \mathrm{otherwise}. \end{array}\right. \end{array}$$



**Proof.** The proof is completed in a manner analogous to the previous lemma by noting that 2∣s+t. Now, let $\left(a,b\right)\in S_{f}\times S_{g}$. From (Equation 5)–(Equation 7),
$$\begin{array}{c} N_{0}=p^{2m-2}+p^{-2}\left(\Lambda_{1}+\Lambda_{2}\right), \end{array}$$
where
$$\begin{array}{cc} \Lambda_{1} & =\left(p-1\right)\varepsilon_{f}\varepsilon_{g}{\sqrt{p}}^{\tau },\\ \Lambda_{2} & =\varepsilon_{f}\varepsilon_{g}{\sqrt{p}}^{\tau }\sum_{z\in F_{p}^{*}}\sigma_{z}\left(\sum_{h\in F_{p}^{*}}\zeta_{p}^{h^{l_{f}}f^{★}\left(a\right)+h^{l_{g}}g^{★}\left(b\right)}\right). \end{array}$$
It is sufficient to determine $\Lambda_{2}$.The first case is that $l_{f}=p-1$.Again, from (Equation 7), we have
$$\begin{array}{cc} \Lambda_{2} & =\frac{p-1}{2}\varepsilon_{f}\varepsilon_{g}{\sqrt{p}}^{\tau }\left(\sum_{z\in F_{p}^{*}}\zeta_{p}^{z(f^{★}\left(a\right)+g^{★}\left(b\right))}+\sum_{z\in F_{p}^{*}}\zeta_{p}^{z(f^{★}\left(a\right)-g^{★}\left(b\right))}\right)\\ \; & =\left\{\begin{array}{cc} {\left(p-1\right)}^{2}\varepsilon_{f}\varepsilon_{g}{\sqrt{p}}^{\tau }, & \mathrm{if}f^{★}\left(a\right)=g^{★}\left(b\right)=0,\\ \frac{p-1}{2}\left(p-2\right)\varepsilon_{f}\varepsilon_{g}{\sqrt{p}}^{\tau }, & \mathrm{if}f^{★}\left(a\right)=-g^{★}\left(b\right)\neq \;0\\ \; & \mathrm{or}f^{★}\left(a\right)=g^{★}\left(b\right)\neq 0,\\ -\left(p-1\right)\varepsilon_{f}\varepsilon_{g}{\sqrt{p}}^{\tau }, & \mathrm{otherwise}. \end{array}\right. \end{array}$$The second case is that $l_{f}=2$ where we only need to consider two different subcases.**Subcase (a)**: If p≡1(mod8), then, from (Equation 7),
$$\begin{array}{cc} \Lambda_{2} & =\varepsilon_{f}\varepsilon_{g}{\sqrt{p}}^{\tau }\sum_{z\in F_{p}^{*}}\sum_{h\in F_{p}^{*}}\zeta_{p}^{z(h^{2}f^{★}\left(a\right)+g^{★}\left(b\right))}\\ \; & =\varepsilon_{f}\varepsilon_{g}{\sqrt{p}}^{\tau }\sum_{z\in F_{p}^{*}}\zeta_{p}^{zg^{★}\left(b\right)}\sum_{h\in F_{p}^{*}}\zeta_{p}^{zh^{2}f^{★}\left(a\right)}\\ \; & =\left\{\begin{array}{cc} {\left(p-1\right)}^{2}\varepsilon_{f}\varepsilon_{g}{\sqrt{p}}^{\tau }, & \mathrm{if}f^{★}\left(a\right)=g^{★}\left(b\right)=0,\\ \left(p+1\right)\varepsilon_{f}\varepsilon_{g}{\sqrt{p}}^{\tau }, & \mathrm{if}f^{★}\left(a\right)g^{★}\left(b\right)\in S_{q},\\ -\left(p-1\right)\varepsilon_{f}\varepsilon_{g}{\sqrt{p}}^{\tau }, & \mathrm{otherwise}. \end{array}\right. \end{array}$$**Subcase (b)**: If p≡5(mod8), then, from (Equation 7),
$$\begin{array}{c} \Lambda_{2}=2\varepsilon_{f}\varepsilon_{g}{\sqrt{p}}^{\tau }\sum_{h\in S_{q}}\sum_{z\in F_{p}^{*}}\zeta_{p}^{z(h^{2}f^{★}\left(a\right)+g^{★}\left(b\right))}. \end{array}$$
The value of $\Lambda_{2}$ is clear if $f^{★}\left(a\right)g^{★}\left(b\right)=0$. We now assume that $f^{★}\left(a\right)g^{★}\left(b\right)\neq 0$. If $\frac{g^{★}\left(b\right)}{f^{★}\left(a\right)}\in C_{2}^{\left(4,p\right)}$; then, the equation $h^{2}f^{★}\left(a\right)+g^{★}\left(b\right)=0$ has exactly two solutions, $h_{1}$ and $h_{2}$, in $S_{q}$, where $h_{2}=-h_{1}$. Hence,
$$\begin{array}{cc} \Lambda_{2} & =2\varepsilon_{f}\varepsilon_{g}{\sqrt{p}}^{\tau }\left(2\left(p-1\right)-\left(\frac{p-1}{2}-2\right)\right)\\ \; & =\left(3p+1\right)\varepsilon_{f}\varepsilon_{g}{\sqrt{p}}^{\tau }. \end{array}$$
Otherwise, if $\frac{g^{★}\left(b\right)}{f^{★}\left(a\right)}\notin C_{2}^{\left(4,p\right)}$, then the inequality $h^{2}f^{★}\left(a\right)+g^{★}\left(b\right)\neq 0$ holds for all *h* in $S_{q}$. Thus,
$$\begin{array}{cc} \Lambda_{2} & =2\varepsilon_{f}\varepsilon_{g}{\sqrt{p}}^{\tau }\times \frac{p-1}{2}\times \left(-1\right)\\ \; & =-\left(p-1\right)\varepsilon_{f}\varepsilon_{g}{\sqrt{p}}^{\tau }. \end{array}$$
So, we conclude that
$$\begin{array}{c} \Lambda_{2}=\left\{\begin{array}{cc} {\left(p-1\right)}^{2}\varepsilon_{f}\varepsilon_{g}{\sqrt{p}}^{\tau }, & \mathrm{if}f^{★}\left(a\right)=g^{★}\left(b\right)=0,\\ \left(3p+1\right)\varepsilon_{f}\varepsilon_{g}{\sqrt{p}}^{\tau }, & \mathrm{if}\frac{g^{★}\left(b\right)}{f^{★}\left(a\right)}\in C_{2}^{\left(4,p\right)},\\ -\left(p-1\right)\varepsilon_{f}\varepsilon_{g}{\sqrt{p}}^{\tau }, & \mathrm{otherwise}. \end{array}\right. \end{array}$$The desired conclusion then follows from (Equation 5), completing the proof. □

**Lemma** **13.**
*Let f,g∈WRPB with $l_{g}=\left(p-1\right)/2$. We suppose that 2∣s+t and (a,b)≠(0,0). We always have $N_{0}=p^{2m-2}$ if $\left(a,b\right)\notin S_{f}\times S_{g}$. Otherwise, the value of $N_{0}$ is presented in the following.*

*When lf=p−1, we have*

$$\begin{array}{c} N_{0}=\left\{\begin{array}{cc} p^{2m-2}+{\left(p-1\right)}^{2}\varepsilon_{f}\varepsilon_{g}{\sqrt{p}}^{\tau -4}, & \mathrm{if}f^{★}\left(a\right)=g^{★}\left(b\right)=0,\\ p^{2m-2}+\frac{p-1}{2}\left(p-2\right)\varepsilon_{f}\varepsilon_{g}{\sqrt{p}}^{\tau -4}, & \mathrm{if}f^{★}\left(a\right)=-g^{★}\left(b\right)\neq \;0\\ \; & \mathrm{or}f^{★}\left(a\right)=g^{★}\left(b\right)\neq 0,\\ p^{2m-2}-\left(p-1\right)\varepsilon_{f}\varepsilon_{g}{\sqrt{p}}^{\tau -4}, & \mathrm{otherwise}. \end{array}\right. \end{array}$$

*When lf=2 and p≡1(mod8), we have*

$$\begin{array}{c} N_{0}=\left\{\begin{array}{cc} p^{2m-2}+{\left(p-1\right)}^{2}\varepsilon_{f}\varepsilon_{g}{\sqrt{p}}^{\tau -4}, & \mathrm{if}f^{★}\left(a\right)=g^{★}\left(b\right)=0,\\ p^{2m-2}+\left(p+1\right)\varepsilon_{f}\varepsilon_{g}{\sqrt{p}}^{\tau -4}, & \mathrm{if}f^{★}\left(a\right)g^{★}\left(b\right)\in S_{q},\\ p^{2m-2}-\left(p-1\right)\varepsilon_{f}\varepsilon_{g}{\sqrt{p}}^{\tau -4}, & \mathrm{otherwise}. \end{array}\right. \end{array}$$

*When lf=2 and p≡5(mod8), we have*

$$\begin{array}{c} N_{0}=\left\{\begin{array}{cc} p^{2m-2}+{\left(p-1\right)}^{2}\varepsilon_{f}\varepsilon_{g}{\sqrt{p}}^{\tau -4}, & \mathrm{if}f^{★}\left(a\right)=g^{★}\left(b\right)=0,\\ p^{2m-2}+\left(3p+1\right)\varepsilon_{f}\varepsilon_{g}{\sqrt{p}}^{\tau -4}, & \mathrm{if}\frac{g^{★}\left(b\right)}{f^{★}\left(a\right)}\in C_{2}^{\left(4,p\right)},\\ p^{2m-2}-\left(p-1\right)\varepsilon_{f}\varepsilon_{g}{\sqrt{p}}^{\tau -4}, & \mathrm{otherwise}. \end{array}\right. \end{array}$$



**Proof.** We note that $\Lambda_{1}=\sum_{z\in F_{p}^{*}}\sigma_{z}\left({\hat{\chi }}_{f}\left(0\right){\hat{\chi }}_{g}\left(0\right)\right)=0$ for f,g∈WRPB. From (Equation 5), $N_{0}=p^{2m-2}+p^{-2}\left(\Lambda_{1}+\Lambda_{2}\right)=p^{2m-2}+p^{-2}\Lambda_{2}$, where $\Lambda_{2}$ is given in Lemma 12. This completes the proof. □

### 4.2. Weight Distributions of $C_{D_{f,g}}$ from WRP or WRPB

The weight distributions of $C_{D_{f,g}}$ defined by (Equation 1) and (Equation 2) are given in the following theorems explicitly. We recall that the length of $C_{D_{f,g}}$, denoted by *n*, is already settled in Lemma 8.

**Theorem** **1.**
*We suppose that 2∤s+t, f,g∈WRP or f,g∈WRPB with $l_{g}=\left(p-1\right)/2$. Then, the code $C_{D_{f,g}}$ has parameters $\left[p^{2m-1}-1,2m\right]$ and its weight distribution is summarized in Table 1 if $l_{f}=p-1$, in Table 2 if $l_{f}=2$ and p≡1(mod8) and in Table 3 if $l_{f}=2$ and p≡5(mod8).*


**Proof.** From Lemma 8, the length is $n=p^{2m-1}-1$. Let (a,b)≠(0,0) and we write wt(c(a,b)) to be the weight of nonzero codewords c(a,b). Clearly,
$$\mathrm{wt}\left(c\left(a,b\right)\right)=n+1-N_{0},$$
where $N_{0}$ is given by Lemma 11. To be more precise, if $\left(a,b\right)\notin S_{f}\times S_{g}$, then
$$\begin{array}{c} \mathrm{wt}\left(c\left(a,b\right)\right)=\left(p-1\right)p^{2m-2}. \end{array}$$For each $\left(a,b\right)\in S_{f}\times S_{g}$, there are four different cases when the weight of c(a,b) does not equal $\left(p-1\right)p^{2m-2}$.When $l_{f}=p-1$, we have
$$\begin{array}{c} \mathrm{wt}\left(c\left(a,b\right)\right)=\left\{\begin{array}{cc} \left(p-1\right)\left(p^{2m-2}-\frac{1}{2}\eta \left(2\right)\varepsilon_{f}\varepsilon_{g}{\sqrt{p}}^{\tau -3}\right), & E_{1}\mathrm{times},\\ \left(p-1\right)\left(p^{2m-2}+\frac{1}{2}\eta \left(2\right)\varepsilon_{f}\varepsilon_{g}{\sqrt{p}}^{\tau -3}\right), & E_{2}\mathrm{times},\\ \left(p-1\right)\left(p^{2m-2}-\varepsilon_{f}\varepsilon_{g}{\sqrt{p}}^{\tau -3}\right), & B_{S_{q}}\mathrm{times},\\ \left(p-1\right)\left(p^{2m-2}+\varepsilon_{f}\varepsilon_{g}{\sqrt{p}}^{\tau -3}\right), & B_{N_{sq}}\mathrm{times}, \end{array}\right. \end{array}$$
where the numbers $B_{S_{q}}$ and $B_{N_{sq}}$ are computed in Lemma 10, and
$$\begin{array}{cc} E_{1} & =\#\left\{\left(a,b\right)\in S_{f}\times S_{g}:f^{★}\left(a\right)\in S_{q},g^{★}\left(b\right)=\pm f^{★}\left(a\right)\right\}=\left(p-1\right)N_{f}\left(i\right)N_{g}\left(i\right),\\ E_{2} & =\#\left\{\left(a,b\right)\in S_{f}\times S_{g}:f^{★}\left(a\right)\in N_{sq},g^{★}\left(b\right)=\pm f^{★}\left(a\right)\right\}=\left(p-1\right)N_{f}\left(j\right)N_{g}\left(j\right), \end{array}$$
with $i\in S_{q}$, $j\in N_{sq}$, and $N_{f}$ and $N_{g}$ are given in Lemma 6. The weight distribution in Table 1 is then established.When $l_{f}=2$ and p≡1(mod8), we have
$$\begin{array}{c} \mathrm{wt}\left(c\left(a,b\right)\right)=\left\{\begin{array}{cc} \left(p-1\right)\left(p^{2m-2}-\varepsilon_{f}\varepsilon_{g}{\sqrt{p}}^{\tau -3}\right), & E_{3}\mathrm{times},\\ \left(p-1\right)\left(p^{2m-2}+\varepsilon_{f}\varepsilon_{g}{\sqrt{p}}^{\tau -3}\right), & E_{4}\mathrm{times},\\ \left(p-1\right)\left(p^{2m-2}+2\varepsilon_{f}\varepsilon_{g}{\sqrt{p}}^{\tau -3}\right), & \frac{{\left(p-1\right)}^{2}}{4}N_{f}\left(i\right)N_{g}\left(i\right)\mathrm{times},\\ \left(p-1\right)\left(p^{2m-2}-2\varepsilon_{f}\varepsilon_{g}{\sqrt{p}}^{\tau -3}\right), & \frac{{\left(p-1\right)}^{2}}{4}N_{f}\left(j\right)N_{g}\left(j\right)\mathrm{times}, \end{array}\right. \end{array}$$
where
$$\begin{array}{cc} E_{3} & =\#\left\{\left(a,b\right)\in S_{f}\times S_{g}:f^{★}\left(a\right)=0,g^{★}\left(b\right)\in S_{q}\right\}+\#\left\{\left(a,b\right)\in S_{f}\times S_{g}:g^{★}\left(b\right)=0,f^{★}\left(a\right)\in S_{q}\right\}\\ \; & =\frac{p-1}{2}\left(N_{f}\left(0\right)N_{g}\left(i\right)+N_{f}\left(i\right)N_{g}\left(0\right)\right),\\ E_{4} & =\#\left\{\left(a,b\right)\in S_{f}\times S_{g}:f^{★}\left(a\right)=0,g^{★}\left(b\right)\in N_{sq}\right\}+\#\left\{\left(a,b\right)\in S_{f}\times S_{g}:g^{★}\left(b\right)=0,f^{★}\left(a\right)\in N_{sq}\right\}\\ \; & =\frac{p-1}{2}\left(N_{f}\left(0\right)N_{g}\left(j\right)+N_{f}\left(j\right)N_{g}\left(0\right)\right), \end{array}$$
for $i\in S_{q}$ and $j\in N_{sq}$. The above argument leads to Table 2.When $l_{f}=2$ and p≡5(mod8), we have
$$\begin{array}{c} \mathrm{wt}\left(c\left(a,b\right)\right)=\left\{\begin{array}{cc} \left(p-1\right)\left(p^{2m-2}-\varepsilon_{f}\varepsilon_{g}{\sqrt{p}}^{\tau -3}\right), & E_{3}\mathrm{times},\\ \left(p-1\right)\left(p^{2m-2}+\varepsilon_{f}\varepsilon_{g}{\sqrt{p}}^{\tau -3}\right), & E_{4}\mathrm{times},\\ \begin{array}{c} c\left(p-1\right)p^{2m-2}-\varepsilon_{f}\varepsilon_{g}{\sqrt{p}}^{\tau -3}\eta \left(u\right)\left(I_{4}\left(\frac{v}{u}\right)-\eta \left(\frac{v}{u}\right)\right)\\ \;\;\mathrm{for}\;\mathrm{all}u,v\in F_{p}^{*}, \end{array} & N_{f}\left(u\right)N_{g}\left(v\right)\mathrm{times}. \end{array}\right. \end{array}$$
The weight distribution in this case is concluded in Table 3. □

**Theorem** **2.**
*We suppose that 2∣s+t and f,g∈WRP with $l_{g}=\left(p-1\right)/2$. Then, $C_{D_{f,g}}$ is an [n,2m] linear code and the weight distribution is given in Table 4 if $l_{f}=p-1$, in Table 5 if $l_{f}=2$ and p≡1(mod8) and in Table 6 if $l_{f}=2$ and p≡5(mod8). Here, we set $n=p^{2m-1}-1+\left(p-1\right)\varepsilon_{f}\varepsilon_{g}{\sqrt{p}}^{\tau -2}$ for brevity.*


**Proof.** The length of this code comes from Lemma 8. For (a,b)≠(0,0), the weight $\mathrm{wt}\left(c\left(a,b\right)\right)=n+1-N_{0}$ can be obtained from Lemma 12. To be more explicit, when $\left(a,b\right)\notin S_{f}\times S_{g}$,
$$\begin{array}{c} \mathrm{wt}\left(c\left(a,b\right)\right)=\left(p-1\right)\left(p^{2m-2}+\left(p-1\right)\varepsilon_{f}\varepsilon_{g}{\sqrt{p}}^{\tau -4}\right). \end{array}$$
The frequency of such codewords equals $p^{2m}-p^{\gamma }$ since f,g∈WRP. When $\left(a,b\right)\in S_{f}\times S_{g}\setminus \left\{\left(0,0\right)\right\}$, we will discuss four different cases.When $l_{f}=p-1$, we have
$$\begin{array}{c} \mathrm{wt}\left(c\left(a,b\right)\right)=\left\{\begin{array}{cc} \left(p-1\right)p^{2m-2}, & N_{f}\left(0\right)N_{g}\left(0\right)-1\mathrm{times},\\ \left(p-1\right)\left(p^{2m-2}+\frac{1}{2}\varepsilon_{f}\varepsilon_{g}{\sqrt{p}}^{\tau -2}\right), & F_{1}\mathrm{times},\\ \left(p-1\right)\left(p^{2m-2}+\varepsilon_{f}\varepsilon_{g}{\sqrt{p}}^{\tau -2}\right), & F_{2}\mathrm{times}, \end{array}\right. \end{array}$$
where we define
$$\begin{array}{cc} F_{1} & =\#\left\{\left(a,b\right)\in S_{f}\times S_{g}:f^{★}\left(a\right)\neq 0,g^{★}\left(b\right)=\pm f^{★}\left(a\right)\right\}=2\sum_{c\in F_{p}^{*}}N_{f}\left(c\right)N_{g}\left(c\right),\\ F_{2} & =p^{\gamma }-N_{f}\left(0\right)N_{g}\left(0\right)-F_{1}. \end{array}$$
Thus, we obtain the weight distribution in Table 4.When $l_{f}=2$ and p≡1(mod8), we have
$$\begin{array}{c} \mathrm{wt}\left(c\left(a,b\right)\right)=\left\{\begin{array}{cc} \left(p-1\right)p^{2m-2}, & N_{f}\left(0\right)N_{g}\left(0\right)-1\mathrm{times},\\ \left(p-1\right)p^{2m-2}+\left(p-3\right)\varepsilon_{f}\varepsilon_{g}{\sqrt{p}}^{\tau -2}, & F_{3}\mathrm{times},\\ \left(p-1\right)\left(p^{2m-2}+\varepsilon_{f}\varepsilon_{g}{\sqrt{p}}^{\tau -2}\right), & F_{4}\mathrm{times}, \end{array}\right. \end{array}$$
where
$$\begin{array}{cc} F_{3} & =\#\left\{\left(a,b\right)\in S_{f}\times S_{g}:f^{★}\left(a\right)g^{★}\left(b\right)\in S_{q}\right\}=\frac{{\left(p-1\right)}^{2}}{4}\left(N_{f}\left(i\right)N_{g}\left(i\right)+N_{f}\left(j\right)N_{g}\left(j\right)\right),\\ F_{4} & =p^{\gamma }-N_{f}\left(0\right)N_{g}\left(0\right)-F_{3}, \end{array}$$
for $i\in S_{q}$ and $j\in N_{sq}$. This implies the weight distribution listed in Table 5.When $l_{f}=2$ and p≡5(mod8), we get
$$\begin{array}{c} \mathrm{wt}\left(c\left(a,b\right)\right)=\left\{\begin{array}{cc} \left(p-1\right)p^{2m-2}, & N_{f}\left(0\right)N_{g}\left(0\right)-1\mathrm{times},\\ \left(p-1\right)p^{2m-2}+\left(p-5\right)\varepsilon_{f}\varepsilon_{g}{\sqrt{p}}^{\tau -2}, & F_{5}\mathrm{times},\\ \left(p-1\right)\left(p^{2m-2}+\varepsilon_{f}\varepsilon_{g}{\sqrt{p}}^{\tau -2}\right), & F_{6}\mathrm{times}, \end{array}\right. \end{array}$$
where we write
$$\begin{array}{cc} F_{5} & =\#\{\left(a,b\right)\in S_{f}\times S_{g}:\frac{g^{★}\left(b\right)}{f^{★}\left(a\right)}\in C_{2}^{\left(4,p\right)}\}\\ \; & =\frac{{\left(p-1\right)}^{2}}{8}\left(N_{f}\left(i\right)N_{g}\left(i\right)+N_{f}\left(j\right)N_{g}\left(j\right)\right)=\frac{1}{2}F_{3},\\ F_{6} & =p^{\gamma }-N_{f}\left(0\right)N_{g}\left(0\right)-\frac{1}{2}F_{3}, \end{array}$$
for $i\in S_{q}$ and $j\in N_{sq}$. Thus, the result in Table 6 is derived. □

**Theorem** **3.**
*We suppose that 2∣s+t and f,g∈WRPB with $l_{g}=\left(p-1\right)/2$. Then, $C_{D_{f,g}}$ is a $\left[p^{2m-1}-1,2m\right]$ linear code with its weight distribution given in Table 7 if $l_{f}=p-1$, in Table 8 if $l_{f}=2$ and p≡1(mod8) and in Table 9 if $l_{f}=2$ and p≡5(mod8).*


**Proof.** We note that (0,0) is not in $S_{f}\times S_{g}$ since f,g∈WRPB. This theorem can be derived in the same way as Theorem 2 by using Lemmas 6–8 and 13. We omitted the details here. □

**Remark** **3.**
*In Theorems 1, 2 and 3, we completely presented the weight distributions of $C_{D_{f,g}}$ for f,g∈WRP or f,g∈WRPB with $l_{f}\in \left\{2,p-1\right\}$ and $l_{g}=\left(p-1\right)/2$, where p≡1(mod4). The case $l_{f}=l_{g}=\left(p-1\right)/2$ is not considered here, since the results for this case will be the same as for $l_{f}=l_{g}=2$ or $l_{f}=l_{g}=p-1$ and they were determined in [17] (see Tables 3, 4 and 6).*


**Remark** **4.**
*For s+t is odd, it is interesting to see that the codes have the same weight distributions whenever the functions are balanced or unbalanced. When s+t is even and f,g∈WRP and p≡1(mod8), the weight distribution in Table 5 coincides with [17] (see Theorem 3.17, Table 5). If we set t=s in Table 5, then the result coincides with [3] (see Theorem 4, Tables 9 and 10). When s+t is even and f,g∈WRPB and p≡1(mod8), the weight distribution in Table 8 coincides with [17] (see Theorem 3.21, Table 7). However, this is not the case for p≡5(mod8). Nevertheless, the index (p−1)/2 is not considered in the literature. Moreover, most of our results, such as Table 1, Table 2, Table 3, Table 4, Table 6, Table 7 and Table 9, are not contained in [3,17].*


Now, we will provide some examples from weakly regular unbalanced plateaued functions to illustrate the results in Theorems 1 and 2.

**Example** **1.**
*Let $f,g:F_{5^{3}}\to F_{5}$ be defined as $f\left(x\right)=\mathrm{Tr}\left(x^{6}+x^{2}\right)$ and $g\left(y\right)=\mathrm{Tr}\left(\theta y^{6}+\theta^{3}y^{2}\right)$ for a primitive element θ of $F_{5^{3}}^{*}$. Then, f,g∈WRP with s=0, t=1, $\varepsilon_{f}=-1$, $\varepsilon_{g}=1$, $l_{f}=l_{g}=2$, ${\hat{\chi }}_{f}\left(\alpha \right)\in \left\{-{\sqrt{5}}^{3}\zeta_{5}^{f^{★}\left(\alpha \right)}\right\}$ and ${\hat{\chi }}_{g}\left(\beta \right)\in \left\{0,5^{2}\zeta_{5}^{g^{★}\left(\beta \right)}\right\}$, where $\alpha ,\beta \in F_{5^{3}}$ and $f^{★}\left(0\right)=g^{★}\left(0\right)=0$. Actually, the function f is quadratic bent and its Walsh transform satisfies $|{\hat{\chi }}_{f}{\left(\alpha \right)|}^{2}=125$. From Magma programs, the code $C_{D_{f,g}}$ is a three-weight code with parameters [3124,6,2400] and the weight enumerator $1+1300z^{2400}+13124z^{2500}+1200z^{2600}$. This is verified by Table 3 in Theorem 1 noting that $I_{4}\left(1\right)=-3$, $I_{4}\left(2\right)=-5$, $I_{4}\left(3\right)=3$ and $I_{4}\left(4\right)=1$.*


**Example** **2.**
*Let $f,g:F_{5^{4}}\to F_{5}$ be defined as $f\left(x\right)=\mathrm{Tr}\left(x^{6}\right)$ and $g\left(y\right)=\mathrm{Tr}\left(y^{26}-y^{2}\right)$. Then, f,g∈WRP with s=t=2, $\varepsilon_{f}=-1$, $\varepsilon_{g}=1$ and $l_{f}=l_{g}=2$. Their Walsh transforms satisfy ${\hat{\chi }}_{f}\left(\alpha \right)\in \left\{0,-5^{3}\zeta_{5}^{f^{★}\left(\alpha \right)}\right\}$ and ${\hat{\chi }}_{g}\left(\beta \right)\in \left\{0,5^{3}\zeta_{5}^{g^{★}\left(\beta \right)}\right\}$, where $\alpha ,\beta \in F_{5^{4}}$ and $f^{★}\left(0\right)=g^{★}\left(0\right)=0$. From Magma programs, the code $C_{D_{f,g}}$ is a three-weight code with parameters [65624,8,50000] and the weight enumerator $1+520z^{50000}+390000z^{52500}+104z^{62500}$. This is verified by Table 6 in Theorem 2.*


**Example** **3.**
*Let $f,g:F_{5^{2}}\to F_{5}$ be defined as $f\left(x\right)=\mathrm{Tr}\left(x^{2}\right)$ and $g\left(y\right)=\mathrm{Tr}\left(\theta y^{2}-\theta y^{6}\right)$ for a primitive element θ of $F_{5^{2}}^{*}$. Then, f,g are quadratic bent functions in the set WRP, with s=t=0, $\varepsilon_{f}=-1$, $\varepsilon_{g}=1$, $l_{f}=l_{g}=2$, ${\hat{\chi }}_{f}\left(\alpha \right)\in \left\{-5\zeta_{5}^{f^{★}\left(\alpha \right)}\right\}$ and ${\hat{\chi }}_{g}\left(\beta \right)\in \left\{5\zeta_{5}^{g^{★}\left(\beta \right)}\right\}$, where $\alpha ,\beta \in F_{5^{2}}$ and $f^{★}\left(0\right)=g^{★}\left(0\right)=0$. From Magma programs, the code $C_{D_{f,g}}$ is a two-weight code with parameters [104,4,80] and the weight enumerator $1+520z^{80}+104z^{100}$. This is also verified by Table 6 in Theorem 2.*


## 5. Minimality of the Codes and Their Applications

This section is devoted to analyzing the minimality of our codes $C_{D_{f,g}}$ defined by (Equation 1) and (Equation 2), and then applying them to construct secret sharing schemes.

A linear code *C* over $F_{p}$ is called minimal if every nonzero codeword c solely covers its scalar multiples zc for $z\in F_{p}^{*}$. In 1998, Ashikhmin and Barg [24] provided a sufficient condition for a linear code to be minimal, that is,
$$\frac{w_{min}}{w_{max}}>\frac{p-1}{p},$$
where $w_{min}$ and $w_{max}$ represent the minimum and maximum nonzero weights, respectively.

Now, we will show the minimality of the constructed linear codes in Theorems 1–3.

**Theorem** **4.**
*(1) The linear codes with weight distributions in Table 1 and Table 2 are minimal, if γ⩾5.*

*(2) The linear codes with weight distributions in Table 4, Table 5 and Table 6 are minimal, if $\varepsilon_{f}\varepsilon_{g}=1$ and γ⩾4, or if $\varepsilon_{f}\varepsilon_{g}=-1$ and γ⩾6.*

*(3) The linear codes with weight distributions in Table 7, Table 8 and Table 9 are minimal, if γ⩾4.*


It should be noted that the minimum distance of $C_{D_{f,g}}^{⊥}$ equals 2 since there are two linearly dependent entries in each codeword in $C_{D_{f,g}}$. Additionally, under the framework stated in [25,26], the minimal codes described in Theorem 4 can be employed to construct secret sharing schemes with good access structure.

**Theorem** **5**(Proposition 2, [26])**.**
*Let C be an [n,k] code over $F_{q}$, and let $G=[g_{0},g_{1},\ldots ,g_{n-1}]$ be its generator matrix. If C is minimal, then in the secret sharing schemes based on the dual code $C^{⊥}$, there are altogether $q^{k-1}$ minimal access sets. In addition, we have the following assertions:*
*(1)* *If $g_{i}$ is a multiple of $g_{0}$, 1⩽i⩽n−1, then participant $P_{i}$ must be in every minimal access set. Such a participant is called a dictatorial participant.**(2)* *If $g_{i}$ is not a multiple of $g_{0}$, 1⩽i⩽n−1, then participant $P_{i}$ must be in $\left(q-1\right)q^{k-2}$ out of $q^{k-1}$ minimal access sets.*

According to Theorem 5, we give the following example for secret sharing schemes.

**Example** **4.***Let $f,g:F_{5^{4}}\to F_{5}$ be defined as $f\left(x\right)=\mathrm{Tr}\left(x^{6}\right)$ and $g\left(y\right)=\mathrm{Tr}\left(y^{6}\right)$. Then, f,g∈WRP with s=t=2, $\varepsilon_{f}=\varepsilon_{g}=-1$ and $l_{f}=l_{g}=2$. From Table 6 in Theorem 2, the code $C_{D_{f,g}}$ is a three-weight code with parameters [90624,8,62500] and the weight enumerator $1+144z^{62500}+390000z^{72500}+480z^{75000}$. So, $C_{D_{f,g}}$ is minimal by Theorem 4. Let $G=[g_{0},g_{1},\ldots ,g_{90623}]$ be the generator matrix of $C_{D_{f,g}}$. Then, in the secret sharing scheme based on the dual code $C_{D_{f,g}}^{⊥}$, there are altogether* 78,125 *minimal access sets. In addition, we have the following assertions:*
*(1)* *If $g_{i}$ is a multiple of $g_{0}$, 1⩽i⩽* 90,623*, then participant $P_{i}$ must be in every minimal access set and $P_{i}$ is a dictatorial participant.**(2)* *If $g_{i}$ is not a multiple of $g_{0}$, 1⩽i⩽* 90,623*, then participant $P_{i}$ must be in* 62,500 *out of* 78,125 *minimal access sets.*


## 6. Conclusions

In the literature, linear codes from weakly regular plateaued functions with index 2 and p−1 have been extensively studied, where *p* is a general prime number, see [3,16,17,18] and the references therein. However, the index of (p−1)/2 has not been considered before. In this paper, we took p≡1(mod4) and studied the construction of new linear codes from two weakly regular plateaued functions with new indexes 2, p−1 and (p−1)/2. By calculating the exponential sums carefully, we succeeded in determining their weight distributions, as we had described in Theorems 1–3. Moreover, most of our codes are minimal and so they are suitable for designing secret sharing schemes. It should be noted that all the examples we gave are chosen from weakly regular unbalanced plateaued functions. Unfortunately, we have not found any weakly regular balanced plateaued functions until now. It would be very nice if someone found such a function in the future.

## Figures and Tables

**Table 1 entropy-26-00455-t001:** The weight distribution of $C_{D_{f,g}}$ in Theorem 1 when $l_{f}=p-1$.

Weight	Frequency
0	1
$\left(p-1\right)p^{2m-2}$	$p^{2m}-1-E_{1}-E_{2}-B_{S_{q}}-B_{N_{sq}}$
$\left(p-1\right)\left(p^{2m-2}-\frac{1}{2}\eta \left(2\right)\varepsilon_{f}\varepsilon_{g}{\sqrt{p}}^{\tau -3}\right)$	$E_{1}$
$\left(p-1\right)\left(p^{2m-2}+\frac{1}{2}\eta \left(2\right)\varepsilon_{f}\varepsilon_{g}{\sqrt{p}}^{\tau -3}\right)$	$E_{2}$
$\left(p-1\right)\left(p^{2m-2}-\varepsilon_{f}\varepsilon_{g}{\sqrt{p}}^{\tau -3}\right)$	$B_{S_{q}}$
$\left(p-1\right)\left(p^{2m-2}+\varepsilon_{f}\varepsilon_{g}{\sqrt{p}}^{\tau -3}\right)$	$B_{N_{sq}}$

**Table 2 entropy-26-00455-t002:** The weight distribution of $C_{D_{f,g}}$ in Theorem 1 when $l_{f}=2$ and p≡1(mod8).

Weight	Frequency
0	1
$\left(p-1\right)p^{2m-2}$	$p^{2m}-1-E_{3}-E_{4}-\frac{{\left(p-1\right)}^{2}}{4}\left(N_{f}\left(i\right)N_{g}\left(i\right)+N_{f}\left(j\right)N_{g}\left(j\right)\right)$
$\left(p-1\right)\left(p^{2m-2}-\varepsilon_{f}\varepsilon_{g}{\sqrt{p}}^{\tau -3}\right)$	$E_{3}$
$\left(p-1\right)\left(p^{2m-2}+\varepsilon_{f}\varepsilon_{g}{\sqrt{p}}^{\tau -3}\right)$	$E_{4}$
$\left(p-1\right)\left(p^{2m-2}+2\varepsilon_{f}\varepsilon_{g}{\sqrt{p}}^{\tau -3}\right)$	$\frac{{\left(p-1\right)}^{2}}{4}N_{f}\left(i\right)N_{g}\left(i\right)$
$\left(p-1\right)\left(p^{2m-2}-2\varepsilon_{f}\varepsilon_{g}{\sqrt{p}}^{\tau -3}\right)$	$\frac{{\left(p-1\right)}^{2}}{4}N_{f}\left(j\right)N_{g}\left(j\right)$

**Table 3 entropy-26-00455-t003:** The weight distribution of $C_{D_{f,g}}$ in Theorem 1 when $l_{f}=2$ and p≡5(mod8).

Weight	Frequency
0	1
$\left(p-1\right)p^{2m-2}$	$p^{2m}-1-E_{3}-E_{4}-\sum_{u,v\in F_{p}^{*}}N_{f}\left(u\right)N_{g}\left(v\right)$
$\left(p-1\right)\left(p^{2m-2}-\varepsilon_{f}\varepsilon_{g}{\sqrt{p}}^{\tau -3}\right)$	$E_{3}$
$\left(p-1\right)\left(p^{2m-2}+\varepsilon_{f}\varepsilon_{g}{\sqrt{p}}^{\tau -3}\right)$	$E_{4}$
$\left(p-1\right)p^{2m-2}-\varepsilon_{f}\varepsilon_{g}{\sqrt{p}}^{\tau -3}\eta \left(u\right)\left(I_{4}\left(\frac{v}{u}\right)-\eta \left(\frac{v}{u}\right)\right)$ for all $u,v\in F_{p}^{*}$	$N_{f}\left(u\right)N_{g}\left(v\right)$

**Table 4 entropy-26-00455-t004:** The weight distribution of $C_{D_{f,g}}$ in Theorem 2 when $l_{f}=p-1$.

Weight	Frequency
0	1
$\left(p-1\right)p^{2m-2}$	$N_{f}\left(0\right)N_{g}\left(0\right)-1$
$\left(p-1\right)\left(p^{2m-2}+\frac{1}{2}\varepsilon_{f}\varepsilon_{g}{\sqrt{p}}^{\tau -2}\right)$	$F_{1}$
$\left(p-1\right)\left(p^{2m-2}+\varepsilon_{f}\varepsilon_{g}{\sqrt{p}}^{\tau -2}\right)$	$p^{\gamma }-N_{f}\left(0\right)N_{g}\left(0\right)-F_{1}$
$\left(p-1\right)\left(p^{2m-2}+\left(p-1\right)\varepsilon_{f}\varepsilon_{g}{\sqrt{p}}^{\tau -4}\right)$	$p^{2m}-p^{\gamma }$

**Table 5 entropy-26-00455-t005:** The weight distribution of $C_{D_{f,g}}$ in Theorem 2 when $l_{f}=2$ and p≡1(mod8).

Weight	Frequency
0	1
$\left(p-1\right)p^{2m-2}$	$N_{f}\left(0\right)N_{g}\left(0\right)-1$
$\left(p-1\right)p^{2m-2}+\left(p-3\right)\varepsilon_{f}\varepsilon_{g}{\sqrt{p}}^{\tau -2}$	$F_{3}$
$\left(p-1\right)\left(p^{2m-2}+\varepsilon_{f}\varepsilon_{g}{\sqrt{p}}^{\tau -2}\right)$	$p^{\gamma }-N_{f}\left(0\right)N_{g}\left(0\right)-F_{3}$
$\left(p-1\right)\left(p^{2m-2}+\left(p-1\right)\varepsilon_{f}\varepsilon_{g}{\sqrt{p}}^{\tau -4}\right)$	$p^{2m}-p^{\gamma }$

**Table 6 entropy-26-00455-t006:** The weight distribution of $C_{D_{f,g}}$ in Theorem 2 when $l_{f}=2$ and p≡5(mod8).

Weight	Frequency
0	1
$\left(p-1\right)p^{2m-2}$	$N_{f}\left(0\right)N_{g}\left(0\right)-1$
$\left(p-1\right)p^{2m-2}+\left(p-5\right)\varepsilon_{f}\varepsilon_{g}{\sqrt{p}}^{\tau -2}$	$\frac{1}{2}F_{3}$
$\left(p-1\right)\left(p^{2m-2}+\varepsilon_{f}\varepsilon_{g}{\sqrt{p}}^{\tau -2}\right)$	$p^{\gamma }-N_{f}\left(0\right)N_{g}\left(0\right)-\frac{1}{2}F_{3}$
$\left(p-1\right)\left(p^{2m-2}+\left(p-1\right)\varepsilon_{f}\varepsilon_{g}{\sqrt{p}}^{\tau -4}\right)$	$p^{2m}-p^{\gamma }$

**Table 7 entropy-26-00455-t007:** The weight distribution of $C_{D_{f,g}}$ in Theorem 3 when $l_{f}=p-1$.

Weight	Frequency
0	1
$\left(p-1\right)\left(p^{2m-2}-\left(p-1\right)\varepsilon_{f}\varepsilon_{g}{\sqrt{p}}^{\tau -4}\right)$	$N_{f}\left(0\right)N_{g}\left(0\right)$
$\left(p-1\right)\left(p^{2m-2}-\frac{p-2}{2}\varepsilon_{f}\varepsilon_{g}{\sqrt{p}}^{\tau -4}\right)$	$F_{1}$
$\left(p-1\right)\left(p^{2m-2}+\varepsilon_{f}\varepsilon_{g}{\sqrt{p}}^{\tau -4}\right)$	$p^{\gamma }-N_{f}\left(0\right)N_{g}\left(0\right)-F_{1}$
$\left(p-1\right)p^{2m-2}$	$p^{2m}-p^{\gamma }-1$

**Table 8 entropy-26-00455-t008:** The weight distribution of $C_{D_{f,g}}$ in Theorem 3 when $l_{f}=2$ and p≡1(mod8).

Weight	Frequency
0	1
$\left(p-1\right)\left(p^{2m-2}-\left(p-1\right)\varepsilon_{f}\varepsilon_{g}{\sqrt{p}}^{\tau -4}\right)$	$N_{f}\left(0\right)N_{g}\left(0\right)$
$\left(p-1\right)p^{2m-2}-\left(p+1\right)\varepsilon_{f}\varepsilon_{g}{\sqrt{p}}^{\tau -4}$	$F_{3}$
$\left(p-1\right)\left(p^{2m-2}+\varepsilon_{f}\varepsilon_{g}{\sqrt{p}}^{\tau -4}\right)$	$p^{\gamma }-N_{f}\left(0\right)N_{g}\left(0\right)-F_{3}$
$\left(p-1\right)p^{2m-2}$	$p^{2m}-p^{\gamma }-1$

**Table 9 entropy-26-00455-t009:** The weight distribution of $C_{D_{f,g}}$ in Theorem 3 when $l_{f}=2$ and p≡5(mod8).

Weight	Frequency
0	1
$\left(p-1\right)\left(p^{2m-2}-\left(p-1\right)\varepsilon_{f}\varepsilon_{g}{\sqrt{p}}^{\tau -4}\right)$	$N_{f}\left(0\right)N_{g}\left(0\right)$
$\left(p-1\right)p^{2m-2}-\left(3p+1\right)\varepsilon_{f}\varepsilon_{g}{\sqrt{p}}^{\tau -4}$	$\frac{1}{2}F_{3}$
$\left(p-1\right)\left(p^{2m-2}+\varepsilon_{f}\varepsilon_{g}{\sqrt{p}}^{\tau -4}\right)$	$p^{\gamma }-N_{f}\left(0\right)N_{g}\left(0\right)-\frac{1}{2}F_{3}$
$\left(p-1\right)p^{2m-2}$	$p^{2m}-p^{\gamma }-1$

## Data Availability

Data is contained within the article.

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
