# Peer review of "Linear Codes Constructed from Two Weakly Regular Plateaued Functions with Index (p − 1)/2"

_entropy, 2024, doi:10.3390/e26060455_

Round 1
Reviewer 1 Report
Comments and Suggestions for Authors
This paper is a continuation of the paper [Yang S. D.; Zhang T. H.; Li P. Linear codes from two weakly regular plateaued balanced functions. Entropy 2023, 25 , 369].
A code is said to be t-weight if the Hamming weights of nonzero codewords can take exactly t different values. t-weight codes have wide applications in theory and practice. For small t, the construction of t-weight codes is challenging. In this paper, an involved methos is used for this. The authors do more: they prove formulas for the weight enumerators. While the formulas are not explicit, their values can be computed for some parameters, yielding new nontrivial examples of t-weight codes. The constructions are based on weakly regular plateaued functions, the proofs use highly nontrivial techniques of exponential sums and Walsh transform. As far as I was able to verify, the results are correct. The paper is well-written, and the language is good.
I strongly recommend the publication of this paper.
Author Response
Dear reviewer 1,
Thanks a lot for your encouragement and your comments. We will modify the language and upload a new version of the paper.
Best,
Shudi Yang
Reviewer 2 Report
Comments and Suggestions for Authors
The article is interesting, but it requires more explanation in order to better undestanding by the reader the magnitude of the presented results. Some comments follow:
- Introduction: An starting paragraph describing the general characteristics and applications of linear codes is desirable.
- The manuscript has several tipos and gramatical errors. As an example, in Line 74 “Plateaued functions in characteristic 2 was firstly initiated by Zheng et al. [ 23]”, it should be “Plateaued functions in characteristic 2 were first studied by Zheng et al. [ 23]
- All sections should include an introductional paragraph. All sections start directly with mathematical definitions or theorems.
- Section 4 should include a paragraph at the end sumarizing the results and their applications
- Section 5 should be extended and the applications of the developed codes have to be described in more detail.
- Section 6:
* The conclusions must be extended, explaining the magnitude of the different results carried out.
* The phrase "Moreover, our codes are suitable for designing secret sharing schemes." must be justified, both in the conclusions and in sections 4 and 5, based on the results presented.
Comments on the Quality of English LanguageThere are some typos and grammatical errors in the manuscript. Some examples are detailed in the comments to authors. Additionally, introductory paragraphs should be included for a better undestanding of the paper.
Reviewer 3 Report
Comments and Suggestions for Authors
Review of: Linear codes constructed from $2$ weakly regular plateaued functions with index $(p-1)/2$:
The authors construct linear codes with only a small number of weights over a field $\mathbb{F}_p$, $p>2$, using/refining a construction from 2007.
As far as I see the results are new (to me) and the proofs (Weil sums) are correct. On the subject there was in Entropy a paper with 3 authors, 2 of them being 2 of the authors of the paper under view. The paper under review seems to be sufficiently different in its aims and results to merit a publication (somewhere). There are several mathematicians interested in this topic and quoted in the paper under review. The paper published on Entropy was quoted in \cite{CSY} which quotes several other older papers (just a quick glance at their Abstracts to see if they merit a quotation).
I would like more details on the secret sharing application mentioned in the last line of Section 5 without quotations (Section 5 of \cite{YT} gave more details).
From the outside it seem that the project started as a 2-name paper (\cite{YT}).
The 3 authors in the Author Contributions make clear the contributions of each author, hence OK for me. But if/when published it is essential that they add in ArXiv a revised 3-named version.
\begin{thebibliography}{99}
\bibitem{CSY} M. \c{C}akmak, A. S{\i}nak, O. Yayla, New self-orthogonal codes from weakly regular plateaued functions and their application in LCD codes,
Crypto e-print 2024-125. ArXiv:2312.04261.
\bibitem{YT} S. Yang and T. Zhang, Linear codes constructed from $2$ weakly regular plateaued functions with index $(p-1)/2$, ArXiv:2303.10833.
\end{thebibliography}

Round 2
Reviewer 2 Report
Comments and Suggestions for Authors
Authors have addressed my suggestions and comments.
Comments on the Quality of English LanguageEnglish is fine, only a final review is required in order to correct some typo.